# Modeling of Severe Plastic Deformation by HSHPT of As-Cast Ti-Nb-Zr-Ta-Fe-O Gum Alloy for Orthopedic Implant

**DOI:** 10.3390/ma16083188

**Published:** 2023-04-18

**Authors:** Dan Cătălin Bîrsan, Carmela Gurău, Florin-Bogdan Marin, Cristian Stefănescu, Gheorghe Gurău

**Affiliations:** Faculty of Engineering, Scholarly Articles for Department of Materials and Environmental Engineering, “Dunărea de Jos” University of Galati, Domnească Street, 47, RO-800008 Galati, Romania

**Keywords:** severe plastic deformation, HSHPT, gum alloy, orthopedic implant, FEA

## Abstract

The High Speed High Pressure Torsion (HSHPT) is the severe plastic deformation method (SPD) designed for the grain refinement of hard-to-deform alloys, and it is able to produce large, rotationally complex shells. In this paper, the new bulk nanostructured Ti-Nb-Zr-Ta-Fe-O Gum metal was investigated using HSHPT. The biomaterial in the as-cast state was simultaneously compressed up to 1 GPa and torsion was applied with friction at a temperature that rose as a pulse in less than 15 s. The interaction between the compression, the torsion, and the intense friction that generates heat requires accurate 3D finite element simulation. Simufact Forming was employed to simulate severe plastic deformation of a shell blank for orthopedic implants using the advancing Patran Tetra elements and adaptable global meshing. The simulation was conducted by applying to the lower anvil a displacement of 4.2 mm in the z-direction and applying a rotational speed of 900 rpm to the upper anvil. The calculations show that the HSHPT accumulated a large plastic deformation strain in a very short time, leading to the desired shape and grain refinement.

## 1. Introduction

Severe plastic deformation (SPD) is a deformation process in large plastic strains that are able to induce in bulk [1] fundamentally different microstructures [2,3] and properties, without changing the chemical composition of the alloys. In particular, SPD leads to a fine or ultrafine structure at room temperature in general, but in some specific SPD processes, this can happen at elevated temperatures as well [4]. An advanced grain refinement determines the outstanding mechanical properties [5]. In specific alloys, the resistance and plasticity, which are otherwise opposed properties, increase simultaneously and at much higher values when the grain size is reduced at the nanoscale [6,7]. The main SPD methods after the number of reported results are Equal Channel Angular Extrusion (ECAE) [8,9,10,11,12], High Pressure Torsion (HPT) [13,14,15,16,17,18,19], and Accumulative Roll Bonding (ARB) [20,21,22,23]. Due to remarkable scientific interest in nanostructured materials that are based on ECAE, HPT, and ARB, more than 59 derived techniques were identified [24]. Among these methods, High Speed High Pressure Torsion is an original one [25] designed to cover materials and shapes that are hard to approach on traditional HPT. High Speed High Pressure Torsion resolves the problem of obtaining ultrafine structures on difficult-to-deform materials like CuAl_13_Ni_4_, Ni_50_Fe_22_Ga_25_Co_3_, or Mg_5.49_Zn_0.55_ Zr. In addition, the technology is able to produce shells and larger disks that cannot be achieved in conventional High Pressure Torsion. Very few papers have reported results on severe-plastic-deformed low modulus Gum metal [26,27,28] and little work has been reported on the optimization of the die design for homogenous strain distribution during deformation, even though researchers have developed different techniques for the severe-plastic-deformation processing of cylindrical shapes. Most of the existing models were developed for the severe plastic deformation of aluminum [29], copper [30], magnesium [31], titanium-based alloys, and steel [32]. The main objective of our research was to predict the metal flow and HSHPT parameters during the SPD of a complex thin shell, starting from the experimental HSHPT of the disk-shaped Ti-Nb-Zr-Ta-Fe-O alloy.

Withal, the purpose of the work was also to examine the possibility of obtaining not only disks but compression staples, which are left blank for HSHPT severe plastic deformation to be applied directly onto the multidirectional compression staples, a technique that provides a nanostructured Ti-Nb-Ta-Zr-O beta-type Gum alloy. In our previous paper [33], we examined the micro-structure refinement of HSHPT disks on the pre-osteoblast response of a newly developed Ti-31.5Nb-3.1Zr-3.1Ta-0.9Fe-0.16O Gum alloy. The ultrafine grain size attained by HSHPT severe plastic deformation conferred the studied GUM alloy an increased biocompatibility that was required for developing promising bone compression staples. As other beta-types, Ti gum alloys exhibit a low modulus of elasticity and multifunctional properties [34,35,36,37,38,39]. Gum metals are high-strength, super-elastic alloys that are based on: (i) alloying elements that stabilize the beta phase (such as Nb, and Ta added with the correct proportion of Zr) and; (ii) the addition of oxygen, which promotes precipitation that acts as strengthener (e.g., NbO_2_) [33]. Moreover, it is well known that a grain size below 1000 nm improves mechanical properties and biological responses compared with their coarse-grained material counterparts [40]. The severely plastic-deformed gum alloy chosen in this study is an ideal candidate for biomedical compression staples due to its remarkable set of mechanical properties and especially its cellular response. Severe plastic deformation is performed to increase the performance of the bio-alloy in terms of its mechanical properties and bone regeneration as a result of a highly refined structure. The ultrafine structure assures a constant compressive load, thus enabling stable bone fixation.

Typically, alloys used for staple applications are stainless steel and super-elastic NiTi alloys [41]. Ti-based alloys possess better biocompatibility than stainless steel and are less allergenic than super-elastic NiTi alloys (caused by nickel ion release). Nickel-free, Ti super-elastic gum alloys may offer a new opportunity for orthopedic implants.

## 2. Materials and Methods

As-cast ingots of Ti-Nb-Zr-Ta-Fe-O gum alloys were produced by levitation melting in a cold crucible MP25 induction furnace (Fives Celes, Lautenbach, France), with the chemical composition given in Table 1. The ingots were machined and cut into disk-shaped samples (20 mm in diameter and 6 mm in thickness).

The HSHPT technique (Figure 1a) is a hot, severe plastic deformation method [16,24,25] that is protected by the patent RO129900. The photos of the genuine HSHPT machine and the die used to fabricate shells are shown in Figure 1e,f. These shells are the starting blank (Figure 1c,d) for implantable multiaxial compression staples for orthopedic surgery (Appendix A). The sample is pressed between two anvils (Figure 1a,b) up to 1 GPa and is simultaneously subjected to torsion by rotation of the upper anvil. If using the traditional HPT method, the sample is torsioned at a very low speed and no slippage (or very well-controlled slippage) between the anvils in the sample occurs, while in the HSHPT method, the superior anvil is rotated at a high speed (×10^2^ rpm).

A great amount of heat is generated by friction in a few seconds and hot severe plastic deformation occurs in the bulk material. This is the key observation showing why this method may process brittle materials in their as-cast states or materials that are very hard to deform.

The recrystallization problem is solved by observing two things: the SPD time is very short and the samples, which are usually thin disks or shells, have a large diameter–thickness ratio. This means that in the HSHPT process, the material is fast-flowing in the cold areas of the anvils, thus losing heat by conduction. The radiation or convection is limited due to fast severe plastic deformation and the low exposed-area outside tools.

## 3. HSHPT Modelling

To perform the severe plastic deformations (SPD), a process simulation, the anvils, and the workpiece were modelled using Inventor software, before being exported as solid geometry files to the Simufact Forming finite element software. The superior anvil and also the inferior one were modelled as rigid bodies. The workpiece was assumed to be a deformable body. Experimental processing data were used to set simulation parameters as follows: a constant inferior punch velocity of 0.525 mm/s and the revolution velocity of the superior punch at 900 rpm. At the beginning of the analysis, all three bodies had been in contact and, at increment zero, the rigid bodies did not have any loads (distributed or point) applied initially.

The determination of contact was done automatically by software according to efficient algorithms. This is a major problem with these types of analyses due to the large number of nodes and elements in the model, and the adaptive meshing process.

### 3.1. Simulation Setup

The model, as shown in Figure 2, presents a regular workpiece, and an upper and lower punch. The initial model consisted of 26,460 tetrahedral elements (type 134—full integration). The workpiece was initially a cylinder shape 5.7 mm high and 15 mm in diameter.

The large strain parameter was included in the “parameter section” to deal with deformations. The deformations and rotations were large enough to invalidate the theory of infinitesimal deformations. In this case, the undeformed and deformed configurations of the model were significantly different, requiring a clear distinction between them. This is usually the case for plastic-deformed materials.

The assumptions assigned for the simulation were the following:Considered material is isotropic;Heat transfer between bodies in this simulation was done by conduction;Heat transfer coefficient between the sample and punch as well as between sample and die was set to 450 W/m^2^/°C;Heat transfer was assumed to occur in all three directions (X, Y, Z);Mechanical and physical material properties (Figure 3) vary with temperature (Young modulus, Poisson coefficient, density and specific heat).The friction conditions between the surfaces of the sample and the dies were considered to be shear with a coefficient of 0.1;

The variation of the Young modulus, the thermal conductivity, and the specific heat capacity with temperature is presented in Table 2.

The three bodies in contact were defined by a Coulomb friction model that was used with a friction coefficient of 0.1. Both anvils were treated as rigid bodies with an initial temperature of 20 °C. The contact table option had been used to indicate that there was no self-contact. The lower punch had a prescribed translated velocity in the negative z-direction of 0.42 mm/s and the revolution velocity of upper punch was 900 rpm.

At the beginning of the analysis, the bodies were in contact, and as the punch was moved in the negative z-direction, it penetrated the deformable sample, changing its shape with the help of the die.

The “adapt global” option was set to indicate that a new mesh was to be generated when the element distortion was based upon a change of strain of 40%. The advancing Patran Tetra elements, with a target number of 10,000 elements, were used.

In manufacturing simulations, the objective is to deform the workpiece from an initial (simple) shape to a final (complex) one. This deformation of the material results in mesh distortion, and for this reason, it is necessary to perform a remeshing step. At this point, a new mesh has been created; the current state of deformation, strains, and stresses has been transferred to the new mesh; and the analysis has been continued.

If a contact tolerance has not been defined, a new one is calculated, based on the new mesh, but this distance can be smaller than the previously calculated tolerance, leading to more iterations.

### 3.2. Theory—Basic Equations

For a compressible, deformable plastic material that meets the yield criterion, we have the equation [42]:(1)$${\bar{\sigma }}^{2}=\frac{1}{2}\left[{\left(\sigma_{x}-\sigma_{y}\right)}^{2}+{\left(\sigma_{y}-\sigma_{z}\right)}^{2}+{\left(\sigma_{z}-\sigma_{x}\right)}^{2}+6\left(\tau_{xy}^{2}+\tau_{yz}^{2}+\tau_{zx}^{2}\right)\right]+k\sigma_{m}^{2}\;{\bar{\sigma }}^{2}=\frac{3}{2}\sigma_{ij}^{\prime }\sigma_{ij}^{\prime }+k\sigma_{m}^{2},$$
where $\sigma_{ij}^{\prime }$ is the deviatoric stress, $\sigma_{m}$ is the hydrostatic stress, and *k* is a constant.

The effective stress depends on the strain rate as can be seen:(2)$$\sigma_{ij}=\frac{\bar{\sigma }}{\dot{\bar{\varepsilon }}}\left[\frac{2}{3}{\dot{\varepsilon }}_{ij}+\left(\frac{1}{k}-\frac{2}{9}\right)\delta_{ij}{\dot{\varepsilon }}_{v}\right],$$
where ${\dot{\varepsilon }}_{v}$ is the volumetric strain rate and $\delta_{ij}=\left\{\begin{array}{l} \begin{array}{ll} 0, & if\;i\neq j \end{array}\\ \begin{array}{ll} 1, & if\;i=j \end{array} \end{array}\right.$.

The equivalent strain rate $\dot{\bar{\varepsilon }}$ is expressed by,
(3)$$\dot{\bar{\varepsilon }}=\sqrt{\frac{2}{9}\left[{\left({\dot{\varepsilon }}_{x}-{\dot{\varepsilon }}_{y}\right)}^{2}+{\left({\dot{\varepsilon }}_{y}-{\dot{\varepsilon }}_{z}\right)}^{2}+{\left({\dot{\varepsilon }}_{z}-{\dot{\varepsilon }}_{x}\right)}^{2}+\frac{3}{2}\left({\dot{\gamma }}_{xy}^{2}+{\dot{\gamma }}_{yz}^{2}+{\dot{\gamma }}_{zx}^{2}\right)\right]+\frac{1}{k}{\dot{\varepsilon }}_{v}^{2}}\;\dot{\bar{\varepsilon }}=\sqrt{\frac{2}{3}{\dot{\varepsilon }}_{ij}^{\prime }{\dot{\varepsilon }}_{ij}^{\prime }+\frac{1}{k}{\dot{\varepsilon }}_{v}^{2}},$$
where ${\dot{\varepsilon }}_{ij}^{\prime }$ is the deviatoric strain rate.

The functional Φ of a strain rate for a sensitive material is given by,
(4)$$\Phi =\int_{V}^{}\left(\int \bar{\sigma }d\dot{\bar{\varepsilon }}\right)dV-\int_{S_{T}}^{}{\bar{F}}_{i}v_{i}dS,$$

${\bar{F}}_{i}$ is the load prescribed on the surface *S_T_*. For a value of $k\in \left(0.0001-0.01\right)$, the volume of material is considered constant because there is a large increase in the functional Φ when the volume changes due to the term of $\frac{1}{k}{\dot{\varepsilon }}_{v}^{2}$ in the equivalent strain rate.

## 4. Results and Discussion

The Simufact Forming post-file contains the results for both the deformable and rigid bodies. When performing a contact analysis, three types of results can be obtained. The first type are the conventional results from the deformable body. This includes the deformation, strains, stresses, and measures of inelastic behaviors such as plastic and creep strains. Besides the reaction forces at conventional boundary conditions, it can be obtained through the contact and friction forces imparted on the body by rigid or other deformable bodies.

Force variation is presented in Figure 4. The experimental plot (Figure 4b) was obtained in the SPD process of the Ti-Nb-Zr-Ta-Fe-O disk shape. The variation shows a plateau at 41.43 kN and a maximum applied force at 451 kN. The HSHPT process lasts roughly 10 s. For the same gum alloy and disk shape sample, the experimental HSHPT parameters are presented in Figure 5.

The variation of the torque, the rotation speed of the upper anvil, and the displacement of the lower anvil confirm that the HSHPT started at 44 s and ended at 51 s. One second after the HSHPT process finished, the measured temperature was 522 °C.

Pressure variation was often used to monitor the evolution of the process. In this case, the severe plastic deformation of the sample occurred at pressures higher than 1 GPa. This can be seen in Figure 6a when the contact pressure exceeds the value of 1 GPa at 2.6 s from the beginning of the simulation and in Figure 6b where the maximum value is reached at the end of the deformation process.

In Figure 7, it can be observed that the temperature of the sample increased during the deformation process at two moments in time. The temperature increases very fast because of the great amount of energy generated in the external friction processes and in the internal friction in the deformed body. The deformed sample transfers heat to the anvils in contact that is exclusively by conduction.

The convection and radiation phenomena are insignificant in the Bridgman cell. The maximum temperature value (542 °C) was validated by the experimental data (Figure 5).

The temperature variation provided valuable information about the progress of severe plastic deformation and it can be used to optimize process parameters to achieve the desired material properties. However, it is important to note that excessive temperatures at the end of HSHPT lead to recrystallization. Therefore, temperature control during the severe plastic deformation process is essential to achieve the desired material properties.

Figure 8 shows Von Mises stress variation in the workpiece that was subjected to SPD at different moments in time. In this way, the distribution of stresses can be observed inside the workpiece but the flow of the deformed material in HSHPT can be observed under the action of the punch (Figure 8d). The variation of Von Mises stress on the upper surface of the workpiece starts at the very beginning of HSHPT, in the 0 to 1 s time interval (Figure 8a). The Von Mises stress increased from 0 to a value of 580 MPa, which shows that the flow limit of the material had been exceeded. The material was reaching the plastic domain at a temperature of 90 °C (Figure 8a,b). By the end of the deformation process, Von Mises stress in the workpiece varied to a small extent in the plastic field, with the maximum value of the equivalent VM stress reaching up to 601 MPa (Figure 8c,d).

Initially, in the early stages of deformation, the Von Mises stress distribution may be relatively uniform on the upper surface of the workpiece. However, as the deformation process continues, the Von Mises stress distribution may become more localized, with areas of high-stress concentration in the regions where the material undergoes the greatest deformation.

In the later stages of the deformation process, the Von Mises stress distribution be-comes more complex, with multiple zones of stress concentration developing on the upper surface of the part. These stress-concentration zones can provide information about the underlying deformation mechanisms, such as the formation of deformation bands or the initiation of shear localization.

Monitoring the Von Mises stress distribution at several points located within the tool radius on the upper surface of the workpiece can provide valuable information about the deformation behavior of the material during severe plastic deformation. This information can be used to optimize process parameters and achieve desired material properties.

The effective plastic strain variation is presented in Figure 9. The inserts show that the distribution of plastic strain in blank shells that underwent HPT at 6 s from the beginning (Figure 9 upper left) to the end of the process (8 s from the beginning—Figure 9 lower right).

The maximum strain predicted by FEA modeling is 3.09. It tends to be the optimum for the shape of the shell blank used for implantable multiaxial compression staples (Figure 9). The complex shape of the orthopedic shell blank imposes this limitation, which, however, is large enough to demonstrate the significant refinement as we have already shown in our previous published paper [33].

Withal, the HSHPT technique combines high hydrostatic compressive stresses with the torsion of the superior punch by rotation, which leads to heating from the friction of the shell. The grain-size refinement takes place in just a few seconds. The finishing temperature of the blanks are well controlled. Consequently, the amorphization is highly improbable.

## 5. Conclusions

Following the modelling of High Speed High Pressure Torsion process performed within the present work, the following can be concluded:A finite element model was developed for the simulation of the severe plastic deformation process based on the following hypotheses: the base material was considered isotropic and its physical characteristics depended on temperature;In the simulation of the deformation process, the total analysis time was considered as the time recorded during the physical HSHPT process;The variations of the Von Mises stress and the total equivalent strain in five points during the deformation process were plotted;The variations of the Von Mises stress and of the deformations along some curves on the surface of the workpiece were drawn at different time intervals.

The FEA result is consistent with the experimental one that indicated the accuracy of the computer simulation. In addition, it shows that the implemented boundary conditions in the simulation were as close as those in experiments.

## Figures and Tables

**Figure 1 materials-16-03188-f001:**
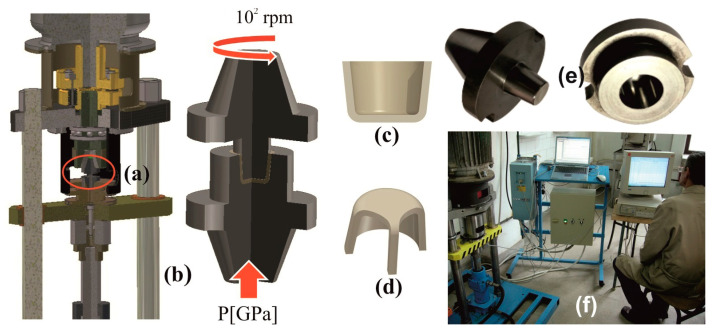
Processing of the blanks by High Speed High Pressure Torsion. (**a**,**b**) HSHPT scheme, (**c**,**d**) compression staples blank, (**e**) HSHPT tools, (**f**) HSHPT experimental setup.

**Figure 2 materials-16-03188-f002:**
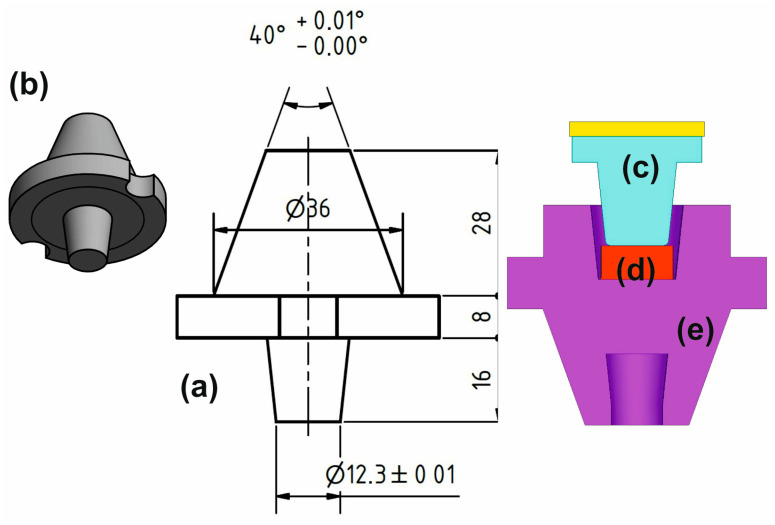
Simulation bodies: (**a**) upper punch geometry, (**b**) 3D upper punch, (**c**) active part of upper punch, (**d**) sample, (**e**) lower punch.

**Figure 3 materials-16-03188-f003:**
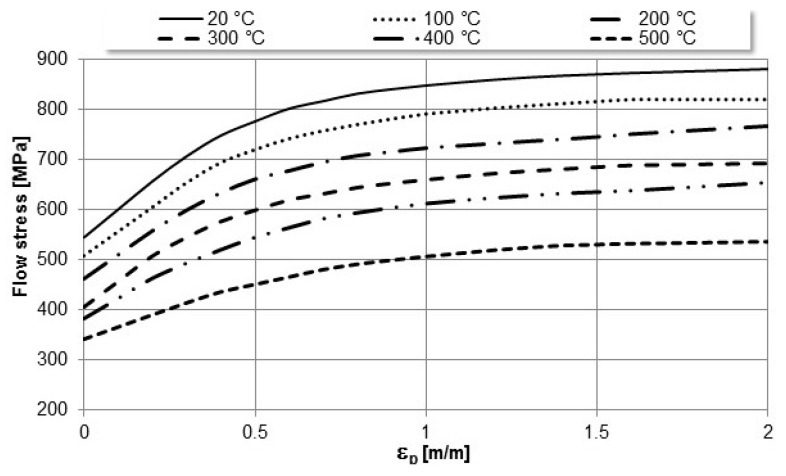
Flow stress for different temperatures.

**Figure 4 materials-16-03188-f004:**
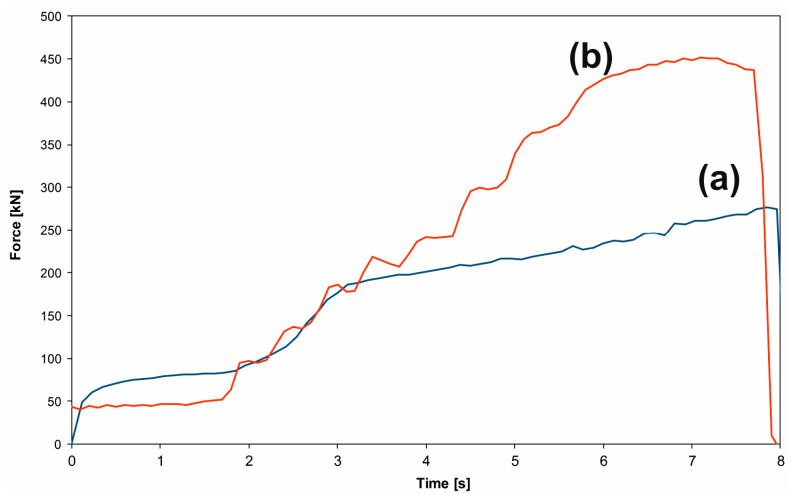
Force variation in the HSHPT process. (a) FEA, (b) experimental for disk-shaped workpiece.

**Figure 5 materials-16-03188-f005:**
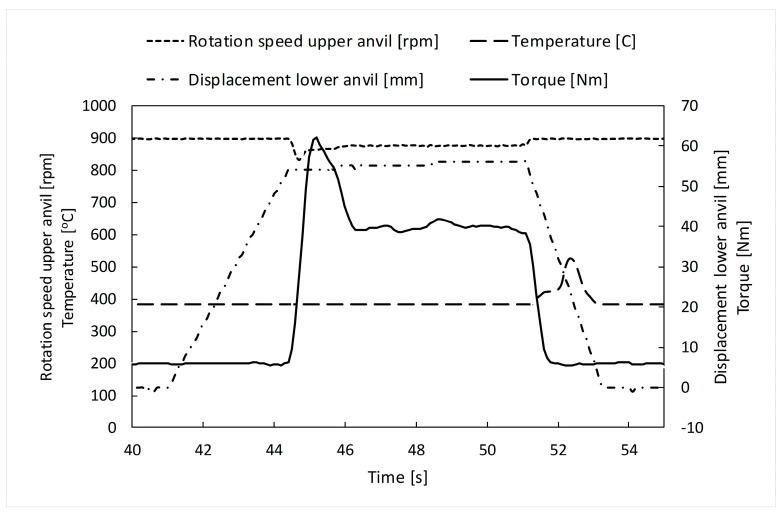
HSHPT experimental parameters.

**Figure 6 materials-16-03188-f006:**
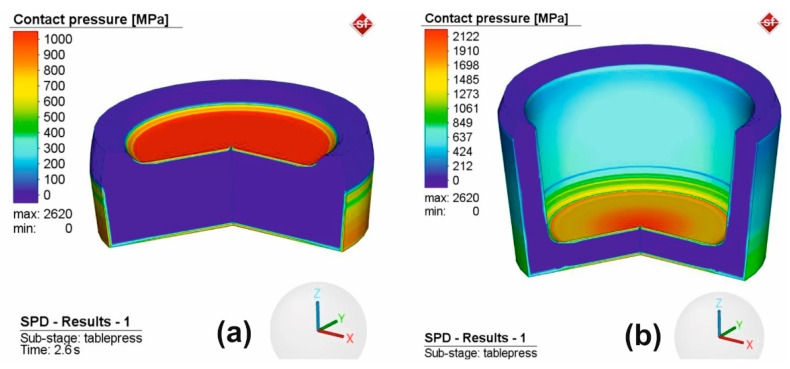
Pressure field: (**a**) at 2.6 s from the SPD beginning and (**b**) at 8 s from the SPD beginning.

**Figure 7 materials-16-03188-f007:**
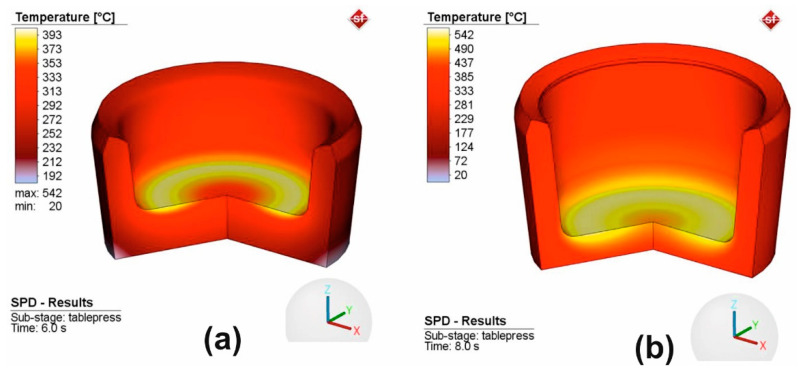
Temperature profile at two moments in time, 6 s (**a**) and 8 s (**b**) from the SPD beginning.

**Figure 8 materials-16-03188-f008:**
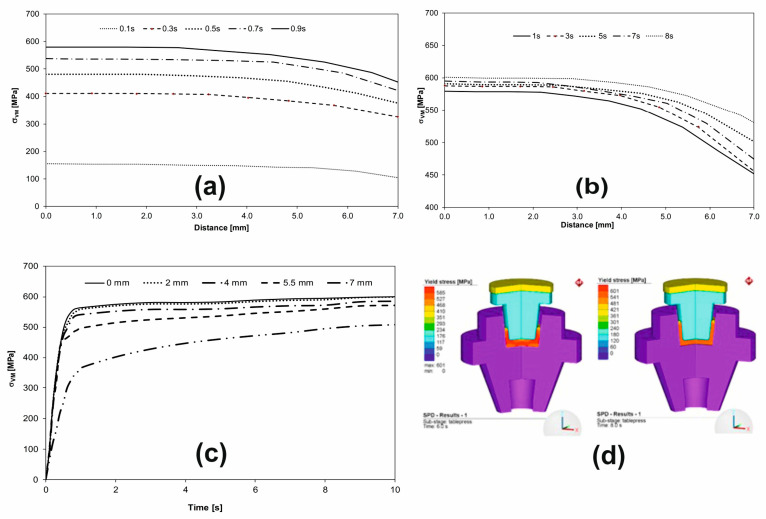
Von Mises stress distribution: (**a**) on the upper surface of the workpiece at the time interval between 0 and 1 s, (**b**) on the upper surface of the workpiece between 1 s and 8 s of the HSHPT process, (**c**) Von Mises stress versus time in the points located on the upper surface of the workpiece, (**d**) Von Mises stresses at two moments of time, 6 s and 8 s.

**Figure 9 materials-16-03188-f009:**
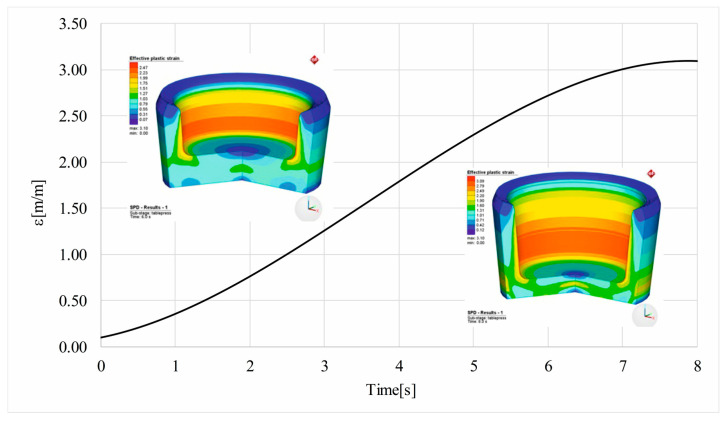
HSHPT strain.

**Table 1 materials-16-03188-t001:** The chemical composition of Ti-Nb-Zr-Ta-Fe-O gum alloy.

Element	Ti	Nb	Zr	Ta	Fe	O
Weight (%)	61.24	31.50	3.10	3.10	0.90	0.16

**Table 2 materials-16-03188-t002:** Variation of the Young modulus, the thermal conductivity and specific heat capacity with temperature.

Temperature [°C]	20	100	200	300	400	500	600
E [GPa]	53.75	53.25	51.75	50	48	45.75	43.25
k [W/(m °C]	26.7	27.7	27.6	27.4	27	26.7	26.4
Cs [J/(kg °C]	501	536	562	554	552	560	607

## Data Availability

Not applicable.

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
