# Peer review of "Modeling of Severe Plastic Deformation by HSHPT of As-Cast Ti-Nb-Zr-Ta-Fe-O Gum Alloy for Orthopedic Implant"

_materials, 2023, doi:10.3390/ma16083188_

Round 1
Reviewer 1 Report
The submitted article in the materials science journal "Materials", in this form is not of interest to readers. The article is missing a lot of research. The article needs radical improvement according to the following remarks.
Comments:
1. At the end of 1 paragraph "Introduction" should indicate the "Purpose of the work"; for what it is necessary to perform this research. Specify the "Objectives of the study." Also, indicate what is the advantage of SPD-treated Ti-Nb-Zr-Ta-Fe-O alloys; why these alloys need to be treated by SPD.
2. In point 2 of the article, please present photos of real HSHPT unit and deformation process of Ti-Nb-Zr-Ta-Fe-O alloy, not 3d models obtained on computer.
3. How does the oxygen content of Ti-Nb-Zr-Ta-Fe-O alloy affect its ability to deform; optimal oxygen content of Ti-Nb-Zr-Ta-Fe-O alloy? Specify in the text of the article.
4. The article is submitted to a materials science journal, the structures of Ti-Nb-Zr-Ta-Fe-O material before and after HSHPT should be presented. What is the effect of grain size in Ti-Nb-Zr-Ta-Fe-O material, after deformation on properties. What is the optimal grain size. Specify in the text of the paper.
5. What is the optimum deformation value of Ti-Nb-Zr-Ta-Fe-O alloy, the structure of the alloy after deformation (according to the optimum parameters). Specify in the text of the article.
6. What happens to the structure and properties of the Ti-Nb-Zr-Ta-Fe-O alloy when the optimum strain is exceeded. Is the amorphization of the alloy structure observed? Specify in the text of the article.
7. Material Science Journal, please present the diffractograms of Ti-Nb-Zr-Ta-Fe-O alloy before and after SPD. Do new phases appear, after deformation?
8. What is the microhardness of the alloy before and after deformation. Specify in the text of the paper.
9. It is necessary to test the materials Ti-Nb-Zr-Ta-Fe-O in tension (construct the tensile curve of the alloy, in the σ-ε axes). Please, present the tensile curves of the alloy before and after deformation.
Author Response
Answers to the comments raised by the Reviewer 1
The authors thank the Reviewer for the competent criticism and constructive comments that improved the manuscript. We have paid close attention to each point raised by the Reviewer and corrected the manuscript by considering his/her comments. The responses are given as follows: the reviewers’ comments in italic font, responses in normal font and changes in the manuscript are in red fonts.
We thank the reviewer for the thorough input on our manuscript and we have addressed all the reviewer’s suggestions as described below.
The submitted article in the materials science journal "Materials", in this form is not of interest to readers. The article is missing a lot of research. The article needs radical improvement according to the following remarks.
Comments:
Q1. At the end of 1 paragraph "Introduction" should indicate the "Purpose of the work"; for what it is necessary to perform this research. Specify the "Objectives of the study." Also, indicate what is the advantage of SPD-treated Ti-Nb-Zr-Ta-Fe-O alloys; why these alloys need to be treated by SPD.
R1: We thank the reviewer for the useful comments. We took your suggestion into consideration and we have indicated more detailed in the manuscript the purpose of the work and the objectives of the study. Also we took your suggestion into consideration and we have provided more information about the severe plastic deformation method used in this experimental study. Please see lanes ….. At the same time, the reference quoted within the text [33] was renumbered since [33] and [34-41] where added.
Withal, the purpose of the work is also to examine the possibility to obtain not only disks but complex shell for multidirectional compression staple directly by HSHPT severe plastic deformation, which provides nanostructured Ti-Nb-Ta-Zr-O beta- type gum alloy. In our previous paper [33] we examine the microstructure refinement of HSHPT disks on the preosteoblast response of a newly developed Ti-31.5Nb-3.1Zr-3.1Ta-0.9Fe-0.16O GUM alloy. The ultrafine grain size attained by HSHPT severe plastic deformation confers the studied GUM alloy increased biocompatibility required for developing promising bone compression staples. As other beta-type Ti gum alloys exhibit low elasticity modulus and multifunctional properties [34-39]. Gum metals are high strength superelastic alloys based on: i) alloying elements that stabilize the beta phase.(such as Nb, Ta added with correct proportion on Zr) and ii) the addition of oxygen that promote precipitation acting as strengthener (e.g. NbO2) [1]. Moreover, it is well known that a grain size below 1000 nm improves mechanical properties and biological response compared to their coarse-grained material counterparts [40]. The severe plastically deformed gum alloy chosen in this study, is an ideal candidate for biomedical compression staples due to its remarkable set of mechanical properties and especially cellular response. Severe plastic deformation is performed to increase the performance of the bioalloy in terms of mechanical properties and bone regeneration as a result of a highly refined structure. The ultrafine structure is assuring constant compressive load enabling stable bone fixation.
Q2. In point 2 of the article, please present photos of real HSHPT unit and deformation process of Ti-Nb-Zr-Ta-Fe-O alloy, not 3d models obtained on computer.
R2. We took the suggestion into consideration and we have added photos of the real HSHPT unit in the supplementary material. Thank you. Please see lines 94 to 97 and the Supplementary Materials Figure S1.
The photos of the genuine HSHPT machine and the die used to fabricate shells are shown in Supplementary Materials Figures S1. These shells are the starting blank (Figure 1. c and d) for implantable multiaxial compression staples for orthopedic surgery (Figure 1.e and Supplementary Materials, Figure S2).
Q3. How does the oxygen content of Ti-Nb-Zr-Ta-Fe-O alloy affect its ability to deform; optimal oxygen content of Ti-Nb-Zr-Ta-Fe-O alloy? Specify in the text of the article.
R3. We thank the reviewer for the useful comments. We took your suggestion into consideration and we have indicated the influence of the oxygen content of Ti gum metals. Please see lines 70 to 74.
Q4. The article is submitted to a materials science journal, the structures of Ti-Nb-Zr-Ta-Fe-O material before and after HSHPT should be presented. What is the effect of grain size in Ti-Nb-Zr-Ta-Fe-O material, after deformation on properties.What is the optimal grain size. Specify in the text of the paper.
R4. We appreciate that reviewer read our work carefully. Indeed, as the reviewer is observing we did not discuss on microstructure, mainly because the main aims of the present paper refer to the FEA model of metal flow in the HSHPT process. The pieces of information on the microstructures of the Ti-Nb-Zr-Ta-Fe-O material before and after HSHPT are presented in our previous paper [33]. There was discussed extensively the microstructure refinement of HSHPT disks.
Please see the reference listed below:
Talling, R.J.; Dashwood, R.J.; Jackson, M.; Dye, D. Compositional variability in gum metal. Scr. Mater. 2009, 60, 1000–1003.
Gurau, C.; Gurau, G.; Mitran, V.; Dan, A.; Cimpean, A. The Influence of Severe Plastic Deformation on Microstructure and In Vitro Biocompatibility of the New Ti-Nb-Zr-Ta-Fe-O Alloy Composition. 2020.
Chen, Q.; Thouas, G.A. Metallic implant biomaterials. Mater. Sci. Eng. R Reports 2015, 87, 1–57.
Q5. What is the optimum deformation value of Ti-Nb-Zr-Ta-Fe-O alloy, the structure of the alloy after deformation (according to the optimum parameters). Specify in the text of the article.
R5. We thank the reviewer for pertinent observation. The maximum strain predicted by FEA modeling is 3.09. It tends to be the optimum for this shape and after our experience it is large enough to determine the fine or ultrafine structure. Please see lines 384-388.
The maximum strain predicted by FEA modeling is 3.09. It tends to be the optimum for the shape of the shell blank used for implantable multiaxial compression staple. The complex shape of orthopedic shell blank impose this limitation which however is large enough to demonstrate significant refinement as we already showed in our previous published paper [33].
Q6. What happens to the structure and properties of the Ti-Nb-Zr-Ta-Fe-O alloy when the optimum strain is exceeded. Is the amorphization of the alloy structure observed? Specify in the text of the article.
R6. Thank you so much for pointing this out. Because the HSHPT is hot SPD technique the amorphization is highly improbable. In the HSHPT process we are controlling parameters to obtain the lowest temperature at the SPD finish to prevent recrystallization (lines 389-392).
Withal, the HSHPT technique combines high hydrostatic compressive stresses with torsion of the superior punch by rotation leading to heating by the friction of the shell. The grain size refinement takes place in just a few seconds. The finishing temperature of the blank are well controlled. Consequently the amorphization is highly improbable.
Q7. Material Science Journal, please present the diffractograms of Ti-Nb-Zr-Ta-Fe-O alloy before and after SPD. Do new phases appear, after deformation?
R7. The pieces of information on the XRD experimental of the Ti-Nb-Zr-Ta-Fe-O material before and after HSHPT are provided in our previous paper [33]. There was examined the influence of HSHPT on XRD spectra.
Q8. What is the microhardness of the alloy before and after deformation. Specify in the text of the paper.
Q9. It is necessary to test the materials Ti-Nb-Zr-Ta-Fe-O in tension (construct the tensile curve of the alloy, in the σ-ε axes). Please, present the tensile curves of the alloy before and after deformation.
R8 and R9. Thank you so much for pointing this out. Our future purpose is to publish another paper approaching the fixation staples manufacturing technology, microstructure and mechanical properties. At this moment the tensile test is performed and the microhardness test is in course.
Thank you very much for your constructive comments.
Kind regards,
Gheorghe Gurău Prof.habil.

Reviewer 2 Report
Study based on SPD by HSHPT for Ti-alloy for orthopaedic implant is highly appreciated. However, the following suggestions and comments to increase the quality of research justification, contributions and findings.
1. In the abstract line 11, “The biomaterial in the as-cast state” instead of “The biomaterial in the as cast state” include the hyphen in between as and caste.
2. In the introduction part, explain why Ti-Nb-Zr-Ta-Fe-O alloy is chosen among all other alloys. May be justify with respect to the biomedical applications.
3. In conventional orthopaedical applications what are the other alloys are used and what techniques used to fabricate the implants. Justify the selection of HSHPT techniques.
4. As mentioned line 27 there will be outstanding mechanical properties due to advanced grain refinement as well as in line 23 and 24 fundamental microstructure changes will have any adverse effect for orthopaedic implants? Or increase the biocompatibility for extra cellular matrix? Justify.
5. Include some more literature in the introduction to find the research gap for the material chose and process identified, in particularly for the orthopaedical implants.
6. Moreover, some introduction about the need for the improvement in the process of orthopaedic implants fabrications need to be justified.
7. Are there any limitations on geometry of the part fabricated using HSHPT and the dimensional accuracy needs to be addressed in the introduction part?
8. In the figure 1 the multidirectional compression staple (e), insert the scale to know the dimension of the component. In the image a label the parts.
9. In line 56 how 1GPa is used to press? any experiments tried before to justify the 1GPa or calibrated through material properties. Is this standard or changes based upon the materials.
10. Justify the speed value? (x102 rpm) Is there will be any significant changes occurs due to the pressing force and speed of the anvil’s values are varied?
11. In line 65 change “as-cast” used hyphen
12. In line 116 It is 900 rpm? Use the same units and notations.
13. In line 161 disc or disk? Use the same spelling in complete manuscript.
14. In figure 5, may be change the graph in different line segment to show the difference instead of color. Because in future readers may take printout in black and white to read the article. In that case the difference identification would be pretty challenging.
15. From figure 6 the mechanical property is determined and figure 7 temperature distributions is determined. Perhaps, the graphs can also be plotted with temperature distribution.
16. In line 193 and 194, does the material is reaching the plastic domain at a temperature of 90 deg C? I hope it should be more than that.
17. What is the scientific inference from the figure 6, 7, and 8? Address them clearly.
18. In the conclusion part, the points should be started with capital letter.
19. Over all, scientific justifications for choosing HSHPT, Materials and results need to be addressed
20. English grammars check is required
Author Response
Answers to the comments raised by the Reviewer 2
The authors thank the Reviewer for the competent criticism and constructive comments that improved the manuscript. We have paid close attention to each point raised by the Reviewer and corrected the manuscript by considering his/her comments. The responses are given as follows: the reviewers’ comments in italic font, responses in normal font and changes in the manuscript are in red fonts.
Study based on SPD by HSHPT for Ti-alloy for orthopedic implant is highly appreciated. However, the following suggestions and comments to increase the quality of research justification, contributions and findings.
R: We thank the reviewer’s appreciation about the subject of our manuscript.
Q1.In the abstract line 11, “The biomaterial in the as-cast state” instead of “The biomaterial in the as cast state” include the hyphen in between as and caste.
R1. The text was modified in accordance with the Reviewer's observation to use “The biomaterial in the as-cast state” instead of “The biomaterial in the as cast state” throughout the manuscript. Thank you.
Q2.In the introduction part, explain why Ti-Nb-Zr-Ta-Fe-O alloy is chosen among all other alloys. May be justify with respect to the biomedical applications.
R2. As the Reviewer 1 also requested, we took this suggestion into consideration and explained why Ti-Nb-Zr-Ta-Fe-O alloy is chosen in this study. Please see lanes 63 to 86.
Q3 In conventional orthopaedical applications what are the other alloys are used and what techniques used to fabricate the implants. Justify the selection of HSHPT techniques.
R3. The compression staples used for biomedical applications are made of austenitic stainless steel or NiTi shape memory alloy. Ti alloys poses higher biocompatibility than all metallic biomaterials. Also beta phase titanium alloys, especially gum metals with low modulus elasticity and multifunctional properties are now ideal candidates for orthopedic implants having excellent osseointegrated response. The motivation for selected the technique used to fabricate the implants was provided in Introduction. Please see lines 74 to 82.
Q4. As mentioned line 27 there will be outstanding mechanical properties due to advanced grain refinement as well as in line 23 and 24 fundamental microstructure changes will have any adverse effect for orthopaedic implants? Or increase the biocompatibility for extra cellular matrix? Justify.
R4. We thank the reviewer for pertinent observation. All required information is in our previous published study [33]. In that paper we studied in vitro biological performance of the Ti-31.5Nb-3.1Zr-3.1Ta-0.9Fe-0.16O GUM alloy subjected to the severe plastic deformation process via HSHPT comparatively with the GUM alloy in the as-cast state. The results were also related to the reference biomaterial, namely commercially pure titanium. The structural investigations showed that by increasing the deformation, a high density of grain boundaries is accumulated, leading gradually to the fine grain size. Cell culture experiments conducted with the MC3T3-E1 cell line proved that all studied surfaces favor the cell adhesion, survival, and proliferation and osteogenic differentiation without showing any deleterious effect. Therefore, the severe plastic deformation by HSHPT technology endows the new GUM alloy with structural features and increased biocompatibility required for developing promising bone compression staples. Please see the lines 75 to 82.
Q5. Include some more literature in the introduction to find the research gap for the material chose and process identified, in particularly for the orthopaedical implants.
R5. The References list was extensively improved according to the Reviewer suggestion (lanes 494-512). Thank you so much for pointing this out.
Q6. Moreover, some introduction about the need for the improvement in the process of orthopaedic implants fabrications need to be justified.
R6. Alloys for medical applications typically require a combination of different properties such as excellent biocompatibility with no adverse tissue reactions and adapted mechanical properties. For use as a bone staple, Young's modulus must be as close as possible to those of bone. The compression staples used for biomedical applications are made of austenitic stainless steel or NiTi shape memory alloy. Stainless steel is a class II short term temporary devices with moderate health risks. Investigation on the biocompatibility of NiTi alloys, reports have been controversial, especially those on orthopedic implant trials. There are serious concerns over the long-term systemic toxicity of nickel ion release. The development of new nickel-free superelastic alloys may offer a new opportunity β microstructure in Ti alloys exhibits a significantly lower modulus than α or α +β microstructure, but there is still a tremendous scope for improvement in terms of alloy design for an ideal orthopedic implant. Among the beta-type Ti-based alloys, the multifunctional Gum Metal possesses “super” properties such as very high strength, low Young's modulus, super elasticity and super plasticity at room temperature. These properties make this alloy an excellent candidate for biomedical sector. Please see lines 83 to 86.
Tipically alloys used for staple application are stainless steel and superelastic NiTi alloy [41]. Ti based alloys possess better biocompatibility then stainless steel and are less allergenic then superelastic NiTi alloy (caused by nickel ion release). Nickel-free, Ti superelastic Gum alloy may offer new opportunity for orthopedic implant.
Q7. Are there any limitations on geometry of the part fabricated using HSHPT and the dimensional accuracy needs to be addressed in the introduction part?
R7. Indeed, as the reviewer is well observing the shape obtained by HSHPT is accurate because high precision of SPD tools but yes there are a few limitations: the dimension of the SPD’ ed blanks are limited at 30-40 mm in diameter and the shells should have rotational symmetry. We have presented all these issues in our previous paper [33].
Please see the reference listed below:
Gurau, C.; Gurau, G.; Mitran, V.; Dan, A.; Cimpean, A. The influence of severe plastic deformation on microstructure and in vitro biocompatibility of the new Ti-Nb-Zr-Ta-Fe-O alloy composition. Materials (Basel). 2020, 13, 1–15.
Q8. In the figure 1 the multidirectional compression staple (e), insert the scale to know the dimension of the component. In the image a label the parts.
R8. The figure 1(e) has been processed again with constructive details of the compression staples and provided in the Supplementary materials (Figure S2).
Q9. In line 56 how 1GPa is used to press? any experiments tried before to justify the 1GPa or calibrated through material properties. Is this standard or changes based upon the materials.
R9.We thank the reviewer for pertinent observation. In the severe plastic deformation process is generally accepted that the 1 GPa pressure level is the minimum pressure necessary for starting of the advanced grain refinement.
Please see the references listed below:
- P.W. Bridgman, On Torsion Combined with Compression, J. Appl. Phys., 1946, 17, p 692–697
- A.P. Zhilyaev and T.G. Langdon, Using High-Pressure Torsion for Metal Processing: Fundamentals and applications, Prog. Mater. Sci.,2008, 53, p 893–979
- R.Z. Valiev, Y. Estrin, Z. Horita, T.G. Langdon, M.J. Zechetbauer, and Y.T. Zhu, Producing bulk ultrafine-grained materials by severe plastic deformation, JOM, 2006, 58(4), p 33–39
- Y. Estrin and A. Vinogradov, Extreme Grain Refinement by Severe Plastic Deformation: A Wealth of Challenging Science, Acta Mater.,2013, 61, p 782–817
Q10. Justify the speed value? (x102 rpm) Is there will be any significant changes occurs due to the pressing force and speed of the anvil’s values are varied?
R10. HSHPT is a patented technique derived from High Pressure Torsion- HPT. In the HPT process, the rotation of the anvil is limited to around one rotation per minute or less no slippage anvil- sample occurs. The key parameter of our method is the rotation speed concurrent with maintaining the same level of the pressure as traditional HPT. The reviewer correctly has observed that the pressure correlated with a certain level of the speed generates by friction high amount of heat in a very short time. That allows us to deform almost any alloy. The reader will find more information in reference [16].
GurÇŽu, G.; GurÇŽu, C.; PotecaÅŸu, O.; Alexandru, P.; Bujoreanu, L.G. Novel high-speed high pressure torsion technology for obtaining Fe-Mn-Si-Cr shape memory alloy active elements. J. Mater. Eng. Perform. 2014, 23, 2396–2402.
Q11.In line 65 change “as-cast” used hyphen
R11. Thank you so much for pointing this out. The corrections have been made.
Q12. In line 116 It is 900 rpm? Use the same units and notations.
Q13. In line 161 disc or disk? Use the same spelling in complete manuscript.
R12 and R13. The text was modified in accordance with the Reviewer's observation to use rpm instead of rot/min and disk instead disc throughout the manuscript. Thank you so much for pointing this out.
Q14. In figure 5, may be change the graph in different line segment to show the difference instead of color. Because in future readers may take printout in black and white to read the article. In that case the difference identification would be pretty challenging.
R14. According to the Reviewer suggestion, the Figures 5 was modified and the plots are shown in a different line segment. Please see line 288.
Q15. From figure 6 the mechanical property is determined and figure 7 temperature distributions is determined. Perhaps, the graphs can also be plotted with temperature distribution.
R15. We thank the reviewer for pertinent observation but HSHPT occurs very fast and the temperature appears like a pulse so we are considering this representation more relevant for reader which could easily observe the temperature distribution in the blank.
Q16. In line 193 and 194, does the material is reaching the plastic domain at a temperature of 90 deg C? I hope it should be more than that.
R16. Because of its high plasticity we succeeded to deform this alloy at room temperature by rolling. In the specific case of HSHPT we consider that 90 Celsius is a reasonable temperature for starting of the metal flow.
Q17. What is the scientific inference from the figure 6, 7, and 8? Address them clearly.
R17. According to the Reviewer’s suggestion, the text was significantly improved. Please see lines 309-310, 335-339, 351-364 and 384 to 388.
Q18. In the conclusion part, the points should be started with capital letter.
R18. Thank you for pointing this out. The text was modified and the points starts now with capital letter.
Q19. Over all, scientific justifications for choosing HSHPT, Materials and results need to be addressed
R19. We thank the reviewer for the thorough input on our manuscript and we have addressed all the reviewer’s suggestions. Please see the lines 63 to 86.
Q20. English grammars check is required
R20. The whole manuscript has been corrected for English errors.
Thank you very much for your constructive comments.
Kind regards,
Gheorghe Gurau Prof.habil.

Reviewer 3 Report
Dear editor
I have read this paper on modeling of severe plastic deformation by HSHPT of as cast Ti-Nb-Zr-Ta-Fe-O. The methodology and experimental procedure as well the FEA simulation were quite interesting. A few points are to be addressed prior to publication of this interesting work.
1. All decimals are to be corrected in Table 1, Figure 2, 3, etc. Comma is to be replaced by the decimal point as is reported in the text too.
2. Table 2, Units column and first column are not aligned correctly.
3. Figure 4, the force variation is not presented in a comparable way. Please re-plot.
4. Figure 8, the internal sketches are not readable. Probable placing them outside the related graphs on the right side is much better.
5. Is is really necessary to repeat the theoretical equations ? They are widely available.
6. Improve the english text as well as the text of the Legends of Figures
Author Response
Answers to the comments raised by the Reviewer 3
The authors thank the Reviewer for the competent criticism and constructive comments that improved the manuscript. We have paid close attention to each point raised by the Reviewer and corrected the manuscript by considering his/her comments. The responses are given as follows: the reviewers’ comments in italic font, responses in normal font and changes in the manuscript are in red fonts.
I have read this paper on modeling of severe plastic deformation by HSHPT of as cast Ti-Nb-Zr-Ta-Fe-O. The methodology and experimental procedure as well the FEA simulation were quite interesting. A few points are to be addressed prior to publication of this interesting work.
R: We thank the reviewer’s appreciation about the subject of our manuscript.
Q1. All decimals are to be corrected in Table 1, Figure 2, 3, etc. Comma is to be replaced by the decimal point as is reported in the text too.
R1. The Table 1 and the Figures 2 and 3 were modified in accordance with the Reviewer's observation to use decimal points instead commas throughout the manuscript. Thank you.
Q2. Table 2, Units column and first column are not aligned correctly.
R2. According to the Reviewer’s suggestion Table 2 was corrected.
Q3. Figure 4, the force variation is not presented in a comparable way. Please re-plot.
R3. According to the Reviewer’s pieces of advice we replotted force variation.
Q4. Figure 8, the internal sketches are not readable. Probable placing them outside the related graphs on the right side is much better.
R4. The Figure 8 inserts have been processed again at full resolution as the reviewer suggested and provided in the Supplementary materials (Figure S3).
Q5. Is is really necessary to repeat the theoretical equations ? They are widely available.
R5.We are grateful to the Reviewer for this suggestion but we consider that some readers will be interested in those equations.
Q6. Improve the english text as well as the text of the Legends of Figures
R6. The whole manuscript has been corrected for English errors.
Thank you very much for your constructive comments.
Kind regards,
Gheorghe Gurău Prof.habil.

Round 2
Reviewer 1 Report
From the presented figure S1, nothing is clear. Figure S1, shows a general view of a non-operating, turned off machine for plastic deformation of a material. It is necessary to submit photographs of the process of plastic deformation of the material, on the switched on, operating installation. Also, add real photos of plastically deformed material and photos of the plastic deformation process on a working installation to the text of the article. Figure 1, which shows 3D models, is replaced with real photos of the process of deformation of materials.
Author Response
Answers to the comments raised by the Reviewer 1
Revision 2
Smilar the Revision 1, the responses are given as follows: the reviewers’ comments in italic font, responses in normal font and changes in the manuscript are highlighted in yellow.
Comments:
Q1. From the presented figure S1, nothing is clear. Figure S1, shows a general view of a non-operating, turned off machine for plastic deformation of a material. It is necessary to submit photographs of the process of plastic deformation of the material, on the switched on, operating installation. Also, add real photos of plastically deformed material and photos of the plastic deformation process on a working installation to the text of the article. Figure 1, which shows 3D models, is replaced with real photos of the process of deformation of materials.
R1: Dear reviewer we fully understand your concern now. From the very beginning of the paper we had honestly presented that this paper shows results of FEA coupled modeling on complex severe plastic deformation process and also complex shape.
In reality we have done this analysis because we need to know the forces, pressures, deformation degrees and metal flow as base for tools design also to know if our HSHPT may support this process. The result was positive and we had designed and machined the tools as you can see in revised figure one.
The next step is to obtain by HSHPT the blanks presented in figure1 c and d (R1). The research is in course as well as the investigations like microhardness and tensile tests.
Before that we had processed implantable compression staples starting from HSHPT’ed disks, stamped and bended, as you can see in images presented below.
Anyway to avoid any confusion we remove from figure 1 the image with the real implant. We just had intended to show how this implant looks like in reality.
The functional HSHPT is also presented in figure 1 f. Please see line 92.
Thank you very much for your constructive comments.
Kind regards,
Gheorghe Gurău Prof.habil.

Reviewer 3 Report
The authors have made significant improvements in their manuscript.
However, corrections, suggested by me in the Figures 4, 8 and 9 were not adopted.
I repeat them in order to help the authors.
1. All axes which present data with decimal numbers must have point (.) not comma (,) in order to denote decimal digits
2. Figure 8, micrographs can be placed adjacent (to the right) of each diagram and then they shall be more visible.
3. Figure 4. Force variation in HSHPT process. (a) FEA, (b) experimental
The (b) experimental data diagram cannot be presented as inset graph. It must be incorporated in the same graph and axes for comparison and discussion purposes.
Please proceed in completing all these amendments.
Author Response
Answers to the comments raised by the Reviewer 3
Revision 2
Similar the Revision 1, the responses are given as follows: the reviewers’ comments in italic font, responses in normal font and changes in the manuscript are highlighted in yellow.
Q The authors have made significant improvements in their manuscript.
- We thank the reviewer’ kind appreciation.
However, corrections, suggested by me in the Figures 4, 8 and 9 were not adopted.
I repeat them in order to help the authors.
Q1. All axes which present data with decimal numbers must have point (.) not comma (,) in order to denote decimal digits
R1. The problem was solved, and yes in our previous revision we lost figures 8a and 8b. Now in this paper, commas were replaced with dots.
Q2. Figure 8, micrographs can be placed adjacent (to the right) of each diagram and then they shall be more visible.
R2. Figure 8 was revised as reviewer suggested. In addition the inserts of figure 8a and 8b were removed because lack of scientific relevance. Please see the line 322. Also the Supplementary material was changed.
Q3. Figure 4. Force variation in HSHPT process. (a) FEA, (b) experimental
The (b) experimental data diagram cannot be presented as inset graph. It must be incorporated in the same graph and axes for comparison and discussion purposes.
Please proceed in completing all these amendments.
R3. The figure 4 was replotted accordingly reviewer suggestion. Please see the line 219.
Kind regards,
Gheorghe Gurău Prof.habil.
